# Fluorescent Platforms for RNA Chemical Biology Research

**DOI:** 10.3390/genes13081348

**Published:** 2022-07-27

**Authors:** Jinxi Du, Ricky Dartawan, William Rice, Forrest Gao, Joseph H. Zhou, Jia Sheng

**Affiliations:** Department of Chemistry, The RNA Institute, University at Albany, State University of New York, 1400 Washington Avenue, Albany, NY 12222, USA; jdu4@albany.edu (J.D.); rdartawan@albany.edu (R.D.); william.rice@stonybrook.edu (W.R.); forrestg357@gmail.com (F.G.); laleezoozoo@gmail.com (J.H.Z.)

**Keywords:** RNA, chemical biology, fluorescent assays

## Abstract

Efficient detection and observation of dynamic RNA changes remain a tremendous challenge. However, the continuous development of fluorescence applications in recent years enhances the efficacy of RNA imaging. Here we summarize some of these developments from different aspects. For example, single-molecule fluorescence in situ hybridization (smFISH) can detect low abundance RNA at the subcellular level. A relatively new aptamer, Mango, is widely applied to label and track RNA activities in living cells. Molecular beacons (MBs) are valid for quantifying both endogenous and exogenous mRNA and microRNA (miRNA). Covalent binding enzyme labeling fluorescent group with RNA of interest (ROI) partially overcomes the RNA length limitation associated with oligonucleotide synthesis. Forced intercalation (FIT) probes are resistant to nuclease degradation upon binding to target RNA and are used to visualize mRNA and messenger ribonucleoprotein (mRNP) activities. We also summarize the importance of some fluorescence spectroscopic techniques in exploring the function and movement of RNA. Single-molecule fluorescence resonance energy transfer (smFRET) has been employed to investigate the dynamic changes of biomolecules by covalently linking biotin to RNA, and a focus on dye selection increases FRET efficiency. Furthermore, the applications of fluorescence assays in drug discovery and drug delivery have been discussed. Fluorescence imaging can also combine with RNA nanotechnology to target tumors. The invention of novel antibacterial drugs targeting non-coding RNAs (ncRNAs) is also possible with steady-state fluorescence-monitored ligand-binding assay and the T-box riboswitch fluorescence anisotropy assay. More recently, COVID-19 tests using fluorescent clustered regularly interspaced short palindromic repeat (CRISPR) technology have been demonstrated to be efficient and clinically useful. In summary, fluorescence assays have significant applications in both fundamental and clinical research and will facilitate the process of RNA-targeted new drug discovery, therefore deserving further development and updating.

## 1. Introduction

Discovered in the 1930s, RNA is an essential nucleic acid molecule that participates in genetic information storage, gene expression, and regulation in living cells [1]. Since then, enormous progress has been made in studying its biological roles in cells, especially over the past four decades, in which scientists have discovered various RNA species and their diverse catalytic functions. Although DNA and RNA are both responsible for regulating gene expression, they have numerous differences. RNA is more vulnerable to degradation than DNA due to the highly reactive hydroxyl group on C2 of the ribose sugar and the existence of ubiquitous ribonucleases (RNases) in cells. RNA also undergoes many dynamic structural changes, and its biological activity can be transient, which make RNA research relatively more challenging. A variety of analytical techniques including fluorescence, bioluminescence, and absorbance-based assays have been developed and demonstrated to be helpful in discovering the functions of mRNA, tRNA, and rRNA in protein synthesis [2], microRNA (miRNA), small interfering RNA (siRNA), and long non-coding RNA (lncRNA) in gene expression and regulation [3], as well as small nuclear RNAs (snRNAs) in RNA splicing and post-transcriptional modifications [4]. Fluorescence and absorbance-based detection have also been instrumental in assessing the concentration and purity of RNA. It is worth noting that although the absorbance technique measures samples in a simple and timely manner, fluorescence displays more significant advantages, including higher sensitivity and higher accuracy. In addition, RNA imaging probes have been widely studied by researchers using both fluorescence and bioluminescence [5]. Bioluminescence is also utilized for RNA detection. It results from a chemical reaction inside an organism, while fluorescence deals with the absorption and emission of lights [6]. Fluorescence offers two main advantages over bioluminescence. First, increased brightness brings out a clearer image of dynamic RNA changes. Second, cofactors and substrates are not required for RNA imaging [7]. These benefits explain why the role of fluorescence in RNA research has received increased attention in recent years. This review will briefly discuss some of these fluorescence techniques and experimental assays, such as fluorescence in situ hybridization, RNA aptamers, molecular beacons, enzymatic labeling, forced intercalation probes, and spectroscopy in studying the conformational changes, metabolic pathways, transportation, and protein interactions of different RNA species. We also discuss their applications in drug development and clinical use.

## 2. Current Fluorescence-Based Experimental Assays and Methods

### 2.1. Application of FISH

Fluorescence in situ hybridization (FISH) was first introduced around 1980 [8]. As the name implies, this method relies on fluorescently labeled DNA or RNA probes which can hybridize with high specificity to some complementary target sequences. These probes are labeled either through radiolabeling, some attached fluorescent protein, or other methods. Because of this, FISH allows for the detection, localization, and even quantification of nucleic acid targets [9]. At the time of its introduction, it was vastly superior to other technologies and pushed the field into a new age of study. Its widespread use has allowed it to mature and become useful for various applications. Along with some current breakthroughs, the technique has been largely improved. For example, various modifications to the probes have allowed for more accessible and more efficient detection of RNA and DNA. Despite these improvements, there are still many challenges in its use in living cells. Nevertheless, the future of FISH could significantly impact the field of medicine by offering live-cell imaging for sick patients.

Currently, it is a vital diagnostic tool for various diseases, particularly tumor growth or cancer. Some cancers can be traced to genetic aberrations or mutations in cellular DNA, which FISH could potentially detect. For example, HER2 is an oncogene associated with breast cancer, coding for a tyrosine kinase that promotes cell proliferation. The amplification of this gene characterizes between 25–30% of breast cancers. Using a probe that targets this oncogene with FISH, physicians can obtain more effective treatments that specifically target the HER2 gene or protein [10].

Initially, FISH required a DNA probe and a target sequence of interest. Since its development, the number of detectable target sequences has increased due to the probes’ combinatorial labeling [11]. The first probes were large as methods of purification and synthesis required greater-sized probes to work. The problem with these large probes was that they were prone to high background fluorescence, which would significantly alter some results. Eventually, probes became much smaller and more efficient. This advanced technology removed the background interference and allowed for more precise results [12]. Also, specific probes such as modified RNA strands were invented, which are used to detect and localize specific RNA targets. These probes are single-stranded RNA used to detect corresponding nucleic acid sequences by hybridization. These probes may be labelled by epitopes, radioisotopes, or fluorophores to highlight target sequences. Combinations of these new technologies have greatly expanded the application of FISH. Combining fluorescence in situ hybridization and high-resolution microscopy has enabled the detection of subcellular localization of RNA. For example, circular RNA (circRNA) is a single-stranded RNA that forms a covalently closed loop; the 3′ and 5′ ends connect to form the circular structure. Using microscopy and FISH to detect circRNA can be challenging since only its junction can be described. circRNAs are expressed at various levels, so to effectively detect these molecules, background fluorescence levels need to be low to obtain precise imaging. To detect these circular strands, high-precision localization using one fluorescently labeled probe spanning the circRNA junction allows for their detection in mammalian cells with high signal-to-noise ratios [13]. In addition, FISH can be used to evaluate mRNA during its complete life cycle, from transcription in the nucleus to maturation and decay in cytoplasm.

Cell heterogeneity cannot be ignored since every cell shows different and independent activities compared to the average performance in a population of cells. Therefore, exploring the detection, quantification, and localization of RNA of a single cell at the subcellular level is necessary. FISH has a low sensitivity and it is almost impossible to use it to monitor a low RNA abundance. A new technique, single-molecule FISH (smFISH) (Figure 1), was invented and has several advantages over normal FISH: (1) FISH typically works to produce qualitative results, while smFISH can quantify the RNA during dynamic gene expression, (2) smFish was designed so that multiple 18–22mer short-stranded DNA probes target the same RNA, significantly increasing the signal–noise ratio by enhancing the fluorescent intensity, thereby lowering the possibility of a biological false positive (BFP), (3) multiple probes binding to the target RNA makes the degradation of a small amount of RNA negligible; this decreases biological false negatives (BFN), (4) smFISH is simple and time efficient; it only requires one-step hybridization, and a single measurement can be realized in 24 h. This technique has been widely employed and has progressed since its birth. For example, single mRNA in *Drosophila melanogaster* could be detected while maintaining the tissue morphology [14]. The optimized smFISH technique was adopted for cryosectioning of zebrafish embryos for cell segmentation and transcription detection [15]. Joshua et al. succeeded in obtaining super-resolution imaging of single mRNA molecule performance in the neural system by combining smFISH with 3D Structured Illumination Microscopy (3D-SIM) [16], and smFish was also used to determine the function of antisense lncRNA transcripts called COOLAIR in silencing *Arabidopsis* FLOWERING LOCUS C (FLC), revealing the mutually exclusive relationship between sense and antisense transcripts at the single-cell level [17]. Overall, smFISH works not only for mRNA and lncRNA localization, but also for RNA migration and RNA–protein interaction. It works not just for mammals, yeasts, and viruses, but also for plants.

Other than smFISH, another significant FISH application worth mentioning is exciton-controlled hybridization-sensitive fluorescent oligonucleotide (ECHO) probes [18] (Figure 2). The labeled RNA samples fluoresce when illuminated by light with specific wavelengths. However, it might not reflect the actual concentration or abundance of RNA due to the high amount of background fluorescent dyes that do not participate in labeling the RNA strand. Researchers must carry out a labor-intensive wash process to remove these excess dyes. One idea to remedy this was to design an on–off switch system, so that when the dye recognizes its target biomolecule, e.g., RNA, the fluorescence is turned on, otherwise, it is turned off. This way, background fluorescence is minimized, as is the need for repetitive washing steps. ECHO probes are thiazole orange-modified fluorescence probes. They show more vigorous fluorescence intensity upon binding to the target RNA, whereas the fluorescence emission in an unbounded state is almost negligible. In this case, the hybridization sensitivity is higher, and background influence was avoided to the greatest extent.

Conventional FISH has difficulty detecting RNA interaction and behaviors in living cells. The ECHO-FISH technology has been adapted to overcome this obstacle. In living cells, the fluorescence intensity of the hybridization complex and the concentration of mRNA have a positive linear correlation because the background is hampered. The photochemical property is tunable and reversible; researchers can incorporate a T-70mer (T70) DNA to pull down the ECHO probe and bind with poly RNA to photo-quench the complex, as displayed by the weak fluorescence intensity. Oomoto et al. used this novel RNA detection technology, ECHO-liveFISH, in 2015 to conduct RNA imaging in living cells by labeling the 28S rRNA and U3 small nucleolar (snoRNA) in mouse brain cells [19]. The ECHO probes used were several 13–50nt oligonucleotide probes with thymine/cytosine residues modified with thiazole orange (TO) dye homodimer. The experiment also proved that the homodimer TO improves the thermal stability of the probe/RNA duplex with increased T_m_.

### 2.2. RNA Aptamer-Based Fluorescent Assays

Aptamers are a new class of synthetic DNA or RNA oligonucleotides that can bind to target molecules with high affinity and specificity. These targets range from nucleic acids, proteins, and organic compounds to cells or even tissues. They are similar in function to antibodies and are thus nicknamed “chemical antibodies” or “nucleic acid antibodies”. However, unlike antibodies and other peptides, they are significantly less likely to produce immune responses. More importantly, they can be chemically modified to change their affinity, specificity, and half-life properties. Even their functions can either be changed or enhanced. This allows one to tailor an aptamer to any specific clinical or investigative need. These are the major advantages that make RNA–aptamers a promising field for investigation in therapeutics and RNA research [20]. Aptamers work by intramolecularly hybridizing into a defined three-dimensional structure that can bind to a target. Many different aptamers have been developed to target RNA. To visualize the location of their targets, aptamers bind fluorophores [21]. Potential aptamers are isolated through an in vitro selection process called systematic evolution of ligands by exponential enrichment (SELEX). The basic idea is that a large pool of oligonucleotides is exposed to a fluorophore, and the small subset of oligos that associate with the fluorophore is then isolated and purified [22].

Notably, several aptamers may bind to and activate fluorophores that mimic the fluorophore found in green fluorescent protein (GFP) [23]. Three of these aptamers are named Spinach, Broccoli, and Pepper. Note that the advantage of light-up aptamers over the traditional GFP technique is that aptamers have a higher signal-to-noise ratio, as in the absence of the aptamer, there is low background fluorescence [24]. Furthermore, aptamers are more specific to RNA To obtain fluorescence, aptamers usually require magnesium to activate; the amount needed depends on the exact aptamer. For example, the Broccoli RNA aptamer requires relatively little magnesium to obtain fluorescence. Figure 3 exhibits how aptamers are lit up.

As one of the most promising aptamers, Mango, a 39mer parallel-stranded G-quadruplex nucleotide, displays several advantages in tracing RNA movement compared to several other methods. Mango’s advantages include but are not limited to high affinity to a fluorescent dye, purification of fluorescently tagged RNA molecules [26], and high specificity and sensitivity for RNA detection [27]. In recent years, it has gradually evolved into a mature background-free detecting technology in living cells. Nested Mango nucleic acid sequence-based amplification (NABSA), as an alternative to RT-PCR, helps detect single-molecule pathogenic RNA [27]. Scientists from Simon Fraser University managed to use this technique to find more detailed information on how SARS-CoV-2 enters and interacts with cells. Also, recently designed second-generation Mango II assay successfully enhanced the imaging contrast of coding and non-coding mCherry RNA at single-molecule resolution without altering their localization. The next-generation Mango aptamer has high affinity, great thermostability, brightness, and improved signal-to-noise ratio compared to other routine aptamers. Therefore, it could be used to measure low abundance RNA, or even a single RNA molecule in the cellular environment. Also, the Mango II assay works well with super-resolution microscopy due to the extended imaging periods, and it works both on fixed and living cells [28]. 

### 2.3. Molecular Beacon-Based Fluorescence Assays

Aside from FISH, researchers managed to design molecular probes called molecular beacons (MBs) that only fluoresce upon binding to target RNA. This way, the signal-to-noise ratio is significantly higher. Molecular beacons are 30–50nt single-stranded oligonucleotide sequences that consist of four components: stem, loop, quencher, and reporter (fluorophore). In its initial state, MBs showed a hairpin stem–loop shape, and the fluorophore was inactivated because the quencher and reporter are too close to each other in an unbounded state. Upon association with the target RNA, the hairpin stem–loop is untied, forming a stable hybrid duplex with the RNA, and the extended spatial conformation activates the fluorophore upon excitation [29]. The technique has another advantage: high specificity in addition to a high signal-to-noise ratio during hybridization and high sensitivity. The hairpin structure can distinguish even one base-pair mismatch while complementary to the target nucleic acid sequences [30]. However, it is challenging and time-consuming to design every MB probe as the stem structure should be neither too strong nor too weak. They hardly hybridize to target sequences if the hairpin structure is too stable. On the other hand, the MBs might spontaneously unfold and activate the fluorophore before binding to the target RNA if the stem base-pair affinity is too weak.

MBs are suitable for measuring RNA trafficking and localization, and their role in the regulation of gene expression [31]. Researchers mainly use this technique to image the endogenous mRNA and miRNA in living cells. However, some barriers need to be overcome. One of these barriers is the efficient delivery of MBs into cells. Because MBs are made of negatively charged RNA/DNA oligonucleotides, the cell membrane is generally impermeable to them, as with many nucleic-acid-based platforms discussed here. Therefore, some MB carriers, such as cationic liposomes, polymers, cell-penetrating peptides (CPPs), and nanoparticles, are designed to accompany the MBs and help them cross the cell membrane [5]. Another challenge is that MBs are prone to giving false positive or inaccurate signals upon entering the cell. Some potential reasons include uneven delivery, sequestration into the nucleus, non-specific binding, and degradation by nucleases.

To overcome these challenges and realize the best accuracy level, Yang et al. invented a new RNA imaging platform named ratiometric bimolecular beacons (RBMBs) [32], which is a hybridization of stem–loop oligonucleotides with a reporter dye on its 5′ end and a linear single-strand nucleotide with a reference dye and quencher on its 5′ and 3′ ends. The reporter dye and quencher are near each other to inactivate the reporter; the quencher and reference dye are far away, so the reference is not influenced. Hybridization separates the reporter and quencher upon binding to the complementary target sequences and fluorescence is observed. The reference dye stays unquenched, and the 3′-UU of the MBs overhang facilitates its nuclear export (Figure 4). The innovative point is incorporating reference dye since heterogeneous delivery of MBs might cause the intervention of background fluorescence. The fluorescence would not only be due to the expression of target RNA but also to the uneven delivery. The incorporation of reference dye minimizes the influence of cell-to-cell variability during MB delivery. Additionally, the incorporation of siRNA elements to the RBMB structure was found to localize the probe in the cytoplasm effectively. The idea is based on recent findings that siRNAs are efficiently transported out of the nucleus by a nuclear transmembrane protein called exportin [33]. The Yang lab also developed a 2′-O-methylated (2′-OMe) RNA with phosphorothioate (PS) inter-nucleotide linkage instead of a phosphodiester (PO) bond [32]. The modifications increase the biostability of the RBMB and add resistance to degradation by nucleases. After several attempts, they found that the 2′-OMe with full PS bond on loop and full PO bond on stem generates the least background fluorescence. Finally, in combination with the RBMBs and smFISH, quantification of RNA transcription at a single molecule level is realizable. Overall, these methods significantly improve the performance of MBs in the intracellular context.

Besides exploring the endogenous RNA, it is also possible to quantify the exogenous miRNA with MBs. A signal-on system that targets and traces dynamic change of miR-124a in neurons was developed recently. The miR-124a beacon showed more intense fluorescence after induction of exogeneous miRNA. It also shows a stronger fluorescence signal after the differentiation of P19 cells, indicating the technique works on measuring both exogenous and endogenous miRNA [34].

The future direction of research with MBs is expected to be a combination of MBs with several known technologies. For example, several nano technologies and MBs can be utilized together. A DNA tetrahedron nanostructure has been designed for MB delivery to detect mRNA levels in tumor cells [35]. This design resists enzyme digestion and delivers the MBs into the cell without carriers. MB coats on molybdenum disulfide (MoS2) nanosheets have been proven to be a versatile probe to detect miRNA after signal amplification by hybridization chain reaction (HCR) technology [36]. Gold nanoparticle (GNP)–nano-MBs can achieve multiple RNA imaging at the same time since multiple MBs tagged with different fluorophore dyes can be loaded onto the GNPs simultaneously [37]. Another application might be amplification prior to the MB’s hybridization to the complementary target RNAs. In a lot of cases, the abundance of mRNA and miRNA is low, and the sensitivity and accuracy are not strong enough to detect and quantify them. Other than the PCR, rolling circle amplification (RCA), signal mediated amplification of RNA technology (SMART), and loop-mediated amplification (LAMP) could also be integrated into MB platforms to provide new possibilities to the applications of traditional MBs.

### 2.4. Enzymatic Labeling-Based Fluorescence Assays

Distinct from other conventional methods, which utilize a noncovalent labeling RNA, chemo-enzymatic labeling strategies use covalent bonding to connect the reporter molecule (fluorophore) with the RNA of interest (ROI). The covalent attachment provides a stronger connection between the probe and ROI. Thus, higher affinity is achieved, making more rigorous purification possible (e.g., washing). High specificity is another advantage of enzymatic labeling over other noncovalent imaging techniques. It is well-known that enzymes target specific RNA structures or sequences, allowing for site-specific labeling of RNA. The chemo-enzymatic method helps incorporate non-natural fluorescent residues into RNA through solid-phase synthesis. RNA reacts with the reporter fluorophore molecule in the secondary chemical step–click reaction after being transfected into the cell. However, synthesis yield becomes much lower if the synthesized RNAs are longer than 50 nt. Enzymatic ligation would be beneficial to overcome the issue of length restriction [38]. Utilizing the chemo-enzymatic method enables improvement of comprehensive kinetics of the reaction, enhancement of the stability of the irreversible fluorescent-tagged RNA, increase in the quantum yield of fluorescence to strengthen the brightness of the RNA to observe the dynamic changes of low abundance RNA, and lastly, optimization of biorthogonal click reactions. Biorthogonal reaction refers to incorporating fluorophore into the transcripts without interfering with biological processes and natural functions. To date, there are a few widely applied click reactions to RNA including: copper(I)-catalyzed azide–alkyne cycloaddition (CuAAC), strain-promoted azide–alkyne cycloaddition (SPAAC), and inverse electron-demand Diels–Alder cycloaddition (IEDDA). The principles of each reaction are shown in Figure 5. Even though CuAAC is the most versatile, it is generally used in fixed and permeable cells in vitro and is not suitable for intracellular environments for three reasons: (1) the toxicity of copper(I) cation might have deleterious effects on RNA function, (2) copper(I) is not stable and is easily oxidized, (3) the Cu(I) oxidation state might accelerate radical-induced RNA degradation.

Both SPAAC and IEDDA strategies are copper-free and show high biocompatibility in vivo. Researchers can use these reactions to add fluorophore reporter molecules onto the target RNA under mild physiological conditions. They are considered promising tools for imaging RNA in living cells. However, the SPAAC reaction is not perfect for all purposes. The cyclooctyne may react with intracellular thiols in vivo as a side reaction, limiting the efficiency of enzymatically transferred modification. The IEDDA reaction also may not work in all types of alkene modifications; it was proved that IEDDA does not react with allyl-modified 5′-cap RNAs. Nevertheless, SPAAC and IEDDA are the most reliable biorthogonal click reactions for imaging cellular RNA because of their outstanding biocompatibility and chemoselectivity to living cells.

Researchers have made incredible breakthroughs in enzymatically labeling RNA. Li et al. managed to use N-(4-aminobutyl)-2-azidoacetamide (AGN) as a substrate and tRNAIle2-agmatidine synthetase (TiaS) as an enzyme to react with BCN-FITC [39]. AGN modified tRNAile2 showed fluorescent labeling as evidenced by acid-urea PAGE, whereas tRNAile2 without AGN showed nothing on PAGE, which means TiaS facilitates the conjugation of AGN and tRNAile2. This indicates that TiaS enables site-specific fluorescent labeling on RNA through click chemistry [39]. Another example of site-specific labeling is the incorporation of a modified nitrogenous substrate PreQ1 into RNA. A 17-nucleoside hairpin RNA ECY-A1 served as the substrate. Given the existence of the enzyme *E. coli* tRNA guanine transglycosylase (TGT), the natural substrate PreQ1 would be substituted to PreQ1 derivative, and the fluorophore dye can be attached to the exocyclic amine of the PreQ1 through a glycol linker. Researchers compared the fluorescent intensity of PreQ1-TO (thiazole orange) before and after incorporation to ECY-A1, and they found that the covalent incorporation leads to a 40-fold increase of fluorescent intensity. They also inserted ECY-A1 into the 3′-UTR of an mRNA transcript coding for mCherry, resulting in a strong fluorescence with the PreQ1-Cy7 and TGT enzyme, while nothing shows up in the absence of TGT [40]. More recently, the same research group optimized the preQ1-TO probe by decreasing the background signal while maintaining high fluorescent intensity; this makes it possible to image mRNA in a fixed cell. Nevertheless, it is still impossible to use TGT in the living cell due to the low cellular concentration and a small range of suitable K_m_ [41].

Other than site-specific labeling, RNA sequence-specific labeling is also possible. The invention of self-alkylating ribozymes uses the small molecule recognition properties of RNA. Unlike linking to a big functional group, small fluorophores are not likely to perturb the biological activities of RNA. In a recent study, an RNA library was built and reacted with selected fluorescein iodoacetamide (FIA); an anti-fluorescein antibody pulled down the sequence, and different ribozyme clone sequences were selected and verified by PAGE followed by fluorescent imaging. The result demonstrates that labeling is specific to the ribozyme sequences, and that the strong background signal from non-specific binding is an obstacle that is difficult to overcome [42]. Holstein et al. showed that the S-adenosyl-L-methionine (AdoMet) enzyme-mediated 4-vinylbenzyl residue can be transferred into 5′-cap of RNA through IEDDA or photo click reactions, and a significant fluorescent increase can be observed after the 4-vinylbenzyl modification has reacted [43].

### 2.5. Forced Intercalation (FIT) Based Fluorescent Probes

Like in ECHO-FISH, forced intercalation (FIT) probes also uses the thiazole orange (TO) family as dyes to dramatically enhance fluorescence upon binding to the target site. It was applied to the investigation of mRNA activities which include, but are not limited to, transcription, splicing, and cytoplasmic localization. A FIT probe is an ssDNA or a single-stranded peptide nucleic acid (ssPNA) containing monomethine cyanine dyes. Unlike FRET or other aptamers, the FIT probe does not require reducing the initial background or separating the signal from the unbounded state [5]. This property essentially makes it a quencher-free probe. The ssDNA/PNA FIT probe emits slight fluorescence in its free state. The cyanine dye intercalates into the sequence upon hybridization with a complementary target mRNA. It serves as a nucleobase surrogate; a significant fluorescence enhancement is observed, allowing easy readout [44]. An ideal FIT probe improves performance in two aspects: brightness and responsiveness [44]. Regarding brightness, the fluorescence enhancement should increase 10- to 100-fold and exclude the intervention from the background. As for high responsiveness, it becomes easy to discriminate the bound and unbound state using the ratios of I_f_/I_0_, where I_f_ is the signal upon hybridization, and I_0_ is the initial fluorescence. The FIT probe should also have a high level of nuclease resistance to avoid degradation upon incorporation with target RNA. As a base surrogate, the dye should show 10- or more-fold fluorescence when hybridized to matched mRNA than the structural disturbance caused by mismatched duplexes (Figure 6). Therefore, the FIT probe is also an effective tool for discriminating the effects of point mutations on DNA/RNA hybridization.

A high extinction coefficient and a high quantum field are needed to improve the performance of brightness. However, self-quenching of TO dye limits the brightness. To facilitate the bathochromic shift, an effort by Hövelmann’s group was made to combine the TO with highly emissive oxazolopyridine analogue JO. The intramolecular dye–dye interaction TO–JO eliminates the excited state when not bound to the target mRNA. Once hybridized, the methine bridge limits the torsion of twisting in a high viscosity environment, the energy transfer between the two dyes is mediated, and the lifetime of fluorescence is enhanced due to the restricted conformational change. They verified this assumption by analyzing the probe with oskar mRNA in *Drosophila* oocytes, combining a DNA-FIT probe with wash-free FISH to track oskar mRNA transport in *Drosophila* oocytes using microinjection [45]. They discovered the TO–JO complex’s brightness is much higher than the TO or JO-only probe at different wavelengths (Figure 7) [46]. This strategy succeeded in localizing oskar mRNA and other poly-A tail-containing mRNA molecules in oocytes from *Drosophila melanogaster*, making the simultaneous localization of multiple mRNAs possible [47]. Two years later, the same research group designed and synthesized an artificial FIT dye named quinoline blue (QB), which is the first chromophore to emit red-light wavelengths because high responsiveness is harder to realize in vivo compared to in vitro due to the autofluorescence. QB is excellent for ruling out background autofluorescence when imaging a living cell. Lastly, they combined QB with other FIT probes, including TO and BO. Hövelmann’s group extended these techniques to visualizing messenger ribonucleoproteins (mRNPs), complexes of mRNA and RNA binding proteins (RBPs) in the living cell. They linked a TO dye as a nucleobase surrogate and used a locked nucleic acid (LNA) adjacent to the dye to afford a brighter fluorescence by making the A-type conformation more rigid, thus increasing the quantum yield of TO emission [46]. 

Later, Kolevzon. et al. designed a PNA-FIT probe that also functions in the red-light range. They synthesized a surrogate base group named BisQ with a monomethine bond similar to TO. The purpose of this design is to discover and complement mutated mRNA, a biomarker in cancer cells. Therefore, the fluorescence enhancement is recorded if BisQ forms complexes with mutated mRNA. Otherwise, there is no significant difference in fluorescence when the probe forms a duplex with unmutated mRNA. They incorporate the PNA-FIT probe into KRAS, a model gene, to track RNA in cancer cell lines. Moreover, it proves that this probe is effective and has potential to diagnose cancer in vivo. Haralampiev et al. used an IAV-QB-FIT probe to complement the 5′ end of a highly conserved domain of small viral RNA (svRNA) and viral RNA (vRNA) of the influenza A virus (IAV) genome [48]. It specifically targets the IAV genome in various species (human, avian, and porcine), and a fluorescent enhancement signal is not detected with influenza B virus. Furthermore, the host cell can also be characterized in different infection states from color signal changes when combining the IAV-QB-FIT probe with an NP-targeting antibody [49]. Overall, the FIT probe is an excellent tool to detect RNA-related diseases and beneficial in exploring single nucleotide polymorphisms (SNPs). FIT also works with aptamers; evidence indicates that aptamers modified with a QB dye upon binding to the target DNA/RNA sequences trigger nontraditional metal-mediated base pair (bp) detection. For example, Hg^2+^ mediates the bridging of thymine and forms T–Hg^2+^–T. The fluorescence is turned on only if the Hg^2+^ is titrated No signal is detected if other metal ions are added in, indicating that this FIT-aptamer is Hg^2+^ sensitive and specific [50].

The main obstacle in using FIT probes to image mRNA in a living cell is cellular delivery. Traditional methods of cellular delivery include microinjection, microporation, and the usage of chemical/biological agents that interact with membranes, such as cell-penetrating peptide (CPP) [51], pore-forming proteins [52], and lipofection [53]. A perturbation probe is also delivered with the assistance of CPPs. However, contrasting toa perturbation probe, an imaging probe must provide precise spatial information, and no localization bias is tolerated. No obvious guideline is made for imaging probe delivery; therefore, some assumptions based on the charges’ influence on the membrane must be made. To overcome this issue, Chamiolo et al. designed an uncharged PNA and negatively charged DNA FIT probe to determine which backbone is better suited for mRNA imaging based on passive diffusion [45]. The results demonstrate that streptolysin-O (SLO), excluding CPPs mediated delivery, can avoid fluorescent spots due to aggregation, and the DNA FIT probe is brighter and more responsive than the PNA-FIT probe [45]. Therefore, the SLO-mediated DNA FIT probe is the best way to image multi-color mRNA accurately.

### 2.6. Fluorescent Techniques for Detection of dsRNA

While the previously discussed methods work reliably for ssRNA, RNA forms a variety of secondary and tertiary structures with double-stranded regions. Although dsRNA detection is relatively understudied, a few previous studies attempted to do so by testing proteins, which could bind to it by selecting for specific secondary or tertiary structural motifs, or oligonucleotides, which could bind to it in a complimentary or antisense fashion. However, formation of the aforementioned structural motifs either slowed down the binding kinetics or produced weak binding interactions in proteins. These same structures are also unable to be recognized by antisense compounds. This meant that detection was limited to the unstructured or partially structured sections of the dsRNA [54]. Another possible method of dsRNA detection relies on the use of oligonucleotides to form RNA triple helixes or triplexes, which arise through tertiary interactions in the major or minor grooves of Watson–Crick base pair stems. This method offers the advantage of detecting specific sequences in dsRNA rather than just structures, but studies in this area have thus far been hindered by weak binding in physiological conditions [54,55]. However, innovations by the labs of Krishna and Sato have allowed for the detection of dsRNA using triplex formation.

This innovation mainly consists of the use of a peptide nucleic acid (PNA). PNAs are similar to oligonucleotides, except the phosphodiester backbone is replaced by a peptide polymer. The advantage of using this compound is that the backbone is neutral, which allows them to have enhanced binding to natural nucleic acids and not interrupt the existing structural motifs. They are also resistant to degradation by both proteases and nucleases [54]. In a similar manner to FIT probes, the Sato lab used an ssPNA modified with thiazole orange (TO) as a base surrogate which would exhibit increased fluorescence upon binding to and intercalation in dsRNA sequences during triplex formation. These triplex-forming PNAs (TFPs) were fittingly called tFIT. They found that their tFIT probes were extremely selective for dsRNA over ssRNA and dsDNA. Furthermore, TO would not fluoresce as strongly if the ssPNA being used did not match the target sequence exactly even by a single base pair, which attests to its extremely high selectivity and therefore, its high-resolution detection abilities. Figure 8 shows a schematic of this binding [55].

Some years later, the Krishna lab took advantage of these properties to detect the viral RNA of the influenza A virus. The dsRNA molecule is an important component of many viruses, either being the viral genome itself or produced in the host cell during infection. In the case of the influenza A, the genome forms a highly conserved dsRNA panhandle structure, which looked to be a promising candidate for targeted detection by a tFIT assay. Instead of TO, they used a modified uracil (U) base which could be detected by fluorescent titration. They found that the fluorescence of their probe increased by four times upon formation of a triplex with the target sequence [54].

## 3. RNA Targeted Fluorescence Spectroscopy

### 3.1. Fluorescence Resonance Energy Transfer (FRET)

The process of FRET involves the non-radiative energy transfer between two chromophores: a donor and an acceptor. These chromophores can be two different photostable dyes, in which one serves as a reporter and the other one as a quencher. Once a photon excites a donor, the donor changes from the ground state to the excited singlet state and transfers some excitation energy to a nearby acceptor upon returning to the ground state. The acceptor is then excited, resulting in fluorescence [56]. During the illumination, the donor’s spectrum must have a smaller wavelength than the acceptor’s absorbance because the initial excited states decay to the state of lower energy quickly due to non-radiative processes such as vibrational relaxation and internal conversion. Consequently, an increased emission intensity is observed on the acceptor’s side [57]. Both wavelengths must have an energy overlap for a successful resonance energy transfer. The more the spectrum overlaps, the higher its FRET efficiency is.

Conventional applications of FRET include, but are not limited to, investigations into RNA–protein interaction, RNA aptamer, and RNA polymerase (RNAP) [33,58,59]. Three crucial components need to be considered to successfully execute the FRET technique to investigate RNA biological activities: the relative distances and the direction between two fluorophores and the concentration of the system. The proper distance should be within 1–10 nm. Side electrical/energy transfer, such as solid coupling between excitons or the tunneling of electrons [60], may occur if the distance is less than 1 nm. If the distance is more than 10 nm, the quantum yield becomes low, energy transfer would be nearly undetectable, and FRET efficiency would be highly impacted (Figure 9). Note that these two fluorophores transfer energy via long-range dipole–dipole interactions. The orientations of these interactive dyes are also of great importance. Investigators may design and attach the dye to a target RNA strand rigidly and adjust the orientation angle κ2 to a higher value to improve the FRET efficiency. The κ2 value is usually between 0 to 4, with 0 meaning that both dyes are perpendicular, 1 meaning that they are parallel, and 4 meaning that the dipoles are collinear. FRET efficiency is also donor–acceptor concentration dependent. In an in vitro experiment, a change in donor or acceptor concentration would quantitatively change the FRET efficiency, and a change in ions would influence its performance as well. Fluorescence fluctuation upon binding state can be mediated by an appropriate concentration of Mg^2+^ and Na^+^. Therefore, investigators should not use the simplified equation, E = R6/(R6 + r6), in which E is the efficiency and R is the Förster distance, when efficiency is 50%, and r is the distances between two fluorophores, to calculate the efficiency of FRET energy transfer since it only accounts for the impact of distances.

In the past decade, the broadest application of FRET in RNA has been the single-molecule FRET study. Researchers tried to observe the dynamic changes of biomolecules during cell processes. They used surface immobilization by covalently linking biotin to RNA. However, the interaction between biotin and RNA may influence RNA’s natural folding, leading to unexpected conformational changes and inaccurate FRET readings. Encapsulating the molecule with a nanoscaled liposome vesicle solved this issue without altering RNA’s functionality. RNA labeled with Cy3/Cy5 dyes and trajectories of their emissions were traced to prove this [61]. In 2017, FRET was utilized to quench the most widely used RNA aptamer, Spinach, through RNA–DNA hybridization. The basic mechanism is to bring the quencher close to it to explore the RNA–protein interaction [60]. FRET is also used to investigate DNA–RNA polymerase complexes during various states to see how DNA bends, extends, and wraps in closed and opened complexes by measuring the distances between Cy3/Cy5 [62]. Thus, FRET is a valuable tool for dynamically exploring the structures of *E. coli* RNAP. Zhao’s group uses fluorescence to label long-strand riboswitches responsible for regulating the btuB gene. Two innovative points of their research are that the fluorescence technique overcomes the issue of labeling RNA strands longer than 200nt, and the selectively binding site is a pair of adjacent adenines. Future research might target cytosine because it includes an exocyclic amino group. The smFRET technique helps visualize and characterize the dynamic conformational equilibrium in the bound state of the riboswitch and its cofactor, adenosylcobalamin [63]. Overall, FRET measurement is beneficial in quantitatively interpreting dynamic changes of RNA conformations and structures.

The most time-consuming step of FRET is the dye selection. Dyes must be photostable, have a high extinction coefficient, and be able to provide high sensitivity. This is why cyanine dyes such as Cy3 and Cy5 are often employed. Researchers have created artificially synthesized dyes to improve the quantum yield and enhance the overall efficiency of FRET. Another difficulty is selecting the specific labeling binding site. Attaching chromophores to inappropriate places may produce high noise signals [64]. FRET is also limited when predicting RNA motion. Lastly, the main parameter it observes is the change of distances between two dyes. It may not be able to monitor the exact movement of the donor and acceptor at any time [64].

### 3.2. Protein-Induced Fluorescence Enhancement (PIFE)

As the term ‘protein-induced fluorescence enhancement’ implies, the fluorophore is not immediately activated upon linking itself onto the RNA. Once it binds to its correlated protein, strong fluorescence is emitted, and a fluorescence intensity histogram is depicted based on thousands of RNA molecule traces during its interaction with proteins (Figure 10). PIFE is an excellent complementary tool to FRET to describe protein–RNA interaction. It records protein movement and activity upon binding to the RNA. Similar to FRET, PIFE can be observed and monitored by total internal reflection fluorescence (TIRF) microscopy. However, it does not simultaneously require two dyes nor the labeling of proteins. Instead, only one dye is needed to label an RNA strand [65]. One benefit of PIFE is that it saves time spent designing and finding the binding site of the dye. It stops any dye from naturally influencing the protein’s binding affinity and dissociation. Typically, the dye is always on the terminal side of the RNA (5′ or 3′) to record and calculate the velocity of motor protein movement along with the RNA strand. The selection of dyes is also akin to FRET with quantum yield, fluorescent intensity, lifetime, dynamic anisotropy, and photostability all taken into account. Cis-trans photoisomerization is a prerequisite of PIFE. The trans state is in the fluorescence excited state, while the cis state is in the nonfluorescent ground state. To extend the lifetime and enhance the intensity, measures should be taken to stabilize the fluorophore into the trans-state.

Taken all together, carbocyanine dye is the most widely used structure at present (e.g., Cy3, Cy5, DY54, etc.) [66]. The carbon–carbon double bonds interconnect two purine rings, and half of the molecule rotates against the other part. It is easy to carry out the cis–trans photoisomerization, and the PIFE effect can be exhibited [67]. The fluorescence intensity is in positive correlation to the surface viscosity. Therefore, if the protein increases the viscosity of RNA, the lifetime and effects of PIFE would be strengthened. Sorokina proved this theory by binding the T7 RNA polymerase to the DNA [68]. In comparison to FRET, PIFE is also distance-dependent and better exhibited in 0–3 nm in vicinity to the fluorophore and protein [66,67]. Researchers used an antivirus protein, RIG-H (truncation mutant of RIG-I) and collected its translocation data on dsRNA. The data indicated that the fluorescence signal intensity weakens from 20–40 bp. However, FRET shows the most sensitivity at 3–10 nm. This offers evidence that FRET and PIFE are good supplements to each other in measuring RNA–protein complexes dynamics for short distances. PIFE is also a good alternative to ITC when calculating the dissociation constant and the Hill coefficient. To fulfill the distance range, PIFE needs a shorter oligonucleotide strand (12mer) than ITC, which usually needs a 21mer strand. The interaction of single-strand RNA binding proteins (SSB), ssRNA, and the Kd can be determined by titrating different concentrations of ssRNA [69]. In addition, PIFE experiments prove that RNA polymerase can be recycled once it finishes transcription and releases the RNA [70]. PIFE should vanish after the polymerase leaves the DNA template after transcription, but instead, the signal occurrence is observed again, and RNAP diffuses in the post-transcription step. The stepwise decay and enhancement signal indicates the direction it diffuses, and the re-initiation of transcription [44]. Lastly, Zhao et al. (2017) offers a new concept, RNA-induced fluorescence enhancement (RIFE), in which the complicated secondary or tertiary structures of RNA motifs might influence the fluorophore’s cis–trans isomerization due to different biomolecular local environments upon binding to the protein [63]. In summary, PIFE can be widely applied to investigate the interaction of RNA with different enzymes, like polymerase, helicase, or ribozyme and riboswitches, to achieve RNA recognition in the future.

### 3.3. Fluorescence Probe in qPCR and RT-PCR

Polymerase chain reaction (PCR) is an outstanding technique widely used in molecular biology to amplify and analyze the DNA and RNA sequences in detail. Compared to other amplification methods, it is cost-effective and can be carried out promptly. However, the ability of PCR is limited when it comes to quantifying the concentration of RNA participating in cell processing, trafficking, and regulation and tracing dynamic changes in every activity to determine the functional complexity of RNA. Because of this, real-time quantitative PCR (qPCR) was invented in the 1990s to overcome these limitations. Although both Northern blotting and qPCR help evaluate gene expression analysis, qPCR has several advantages compared to Northern blotting, as it is more time efficient, has more precision (sub microgram level), and is less tedious than Northern blotting [71]. To quantify the total RNA, we need to set an arbitrary fluorescence threshold and the total number of cycles required to exceed it, also known as quantitative cycle (Cq). The more DNA presented before carrying out qPCR, the less Cq we need to surpass the threshold. We may then use a standard serial solution with different known nucleic acid concentrations to draw a calibration curve in the form of Cq over-concentration (Y vs. X). We can quantify the unknown RNA concentration by measuring the Cq once we incorporate the data into the curve. During the dynamic change, we can use a fluorescence oligonucleotide hydrolysis probe to monitor the increase of the PCR product. Hydrolysis probes are widely used in several medical conditions, such as genotyping, pharmacogenomics, or diagnostics. In detail, the fluorophore is on the 5′-end, and the quencher is on the 3′-end, with fluorescence extinguished due to the oligonucleotide being intact. During the annealing step, it binds to the amplified region, and during the extension of qPCR reaction, the 5′ reporter is cleaved and activated, emitting fluorescence (Figure 11). This method is useful for quantifying and visualizing mRNA and provides possibilities for designing new biochips or biosensors.

### 3.4. Fluorescence-Based Assays in Determining RNA-Protein Binding Sites

The study of RNA–protein interaction is prevalent in the present RNA biology. We can also use it in the clinical environment. For instance, if we find the position of a single-stranded RNA binding site in cancer-related proteins, we can synthesize an artificial analog to bind with the natural RNA. The translation could be suppressed, and the diseases might be controlled. For example, the SARS-CoV-2 viral RNA is known to bind with ACE2 protein. An updated research about COVID-19 suggests that scientists could synthesize part of the ACE2 receptor to bind with viral RNA [72]. The α-helix peptide, consisting of 23 amino acids, specifically binds with viral RNA to prevent it from entering the human cells to limit the translation and expression and shows higher affinity than ACE2. This research shows the significance of finding the RNA–protein binding sites, and the affinity indicates the level of stability of corresponding complexes. There are three ways to image the binding site: X-ray crystallography, NMR, and fluorescence. The advantage of fluorescence over X-ray crystallography is that it overcomes the phase issue. NMR does not apply to the circumstance of big RNA–protein complexes. There are two methods available currently. First, we can measure the fluorescence anisotropy/polarization of fluorophore-labeled RNA to characterize the binding event. Secondly, researchers may measure the difference between the fluorescence change before and after the RNA–protein binding. After we find the affinity of RNA–protein complexes, the equilibrium dissociation constant (K_d_) is obtained, which indicates the stability of RNA–protein complexes in the solution state.

## 4. Application in RNA-Based Drug Delivery and Discovery

### 4.1. Fluorescence Techniques in RNA Therapeutics

Fluorescence assays have made a big impact on medicinal chemistry. For example, they have aided in studies on RNA drug delivery. Novel RNA drug therapies are appealing but have low uptake efficiencies into cells. Researchers believed that natural mediums, instead of the artificial ones, could be an effective tool to carry out RNA drug delivery as they would eliminate the need to adjust the medium to a pseudo-environment. Specifically, extracellular vesicles (EVs) are the natural mode of RNA exchange in eukaryotic cells and have been recognized as good drug delivery vehicles. Jong et al. observed the process of fluorescently labeled EVs entering HeLa cells and deduced that EV uptake is primarily due to clathrin-independent endocytosis rather than clathrin-mediated endocytosis [73]. Though EV shows excellent uptake efficiency, it is impossible to implement batch production. Scientists discovered that human red blood cell (RBC) EV can be adopted as a strong candidate to resolve this dilemma. It could be gained because of two main reasons: first, RBCs are readily available as they are numerous in the body; secondly, RBCs do not have any nuclear or mitochondrial DNA, thus reducing the risk of horizontal gene transfer. In an experiment by Usman et al., FAM and DiR fluorescence was observed and recorded to see if the fluorescently labeled RBC successfully delivered the antisense oligonucleotide (AEOs) into leukemia and breast cancer cells. The location and intensity of fluorescence indicate that the RBCEV can deliver the drugs into the cell without cytotoxicity [74]. Another study on RNA drug delivery which also used FAM discusses an RNA–triple-helix hydrogel, which comprises specific miRNA molecules and targets human cancers. FAM was used to label the miRNA-205 (tumor suppressor) and miRNA-221 (oncomiR inhibitor) of synthetically designed RNA–triple-helix hydrogel, and the relative fluorescence intensities observed were used to verify the proper formation of the triple helix, as well as determine the optimal concentration of the miRNA [75]. Fluorescence imaging techniques also plays a significant role in delivering RNA nanoparticle therapeutics to brain tumors. Researchers designed a packing RNA (pRNA) of bacteriophage phi29 DNA to self-construct mature RNA nanoparticles, including therapeutic agents, targeting ligand/aptamer and fluorophore modules. Monitoring the delivery of the pRNA-3WJ motif with an imaging technique would be beneficial in developing other multiuse RNA nanoparticles that disrupt the pathway of pathogenesis [76].

The drug delivery fluorescence technique plays an important role in designing and discovering new drugs. A screening cascade targeting the T-box riboswitch anti-terminator element was invented to investigate new anti-bacterial drugs. The T-box riboswitch was found in gram-positive bacteria. It is located upstream to the mRNA that codes for the aminoacyl–tRNA synthetases and regulates transcription by binding or unbinding to tRNA. The stem–loop I of the T-box leader forms a complex with uncharged tRNA, and if tRNA does not form a complex with the amino acid residue, an antitermination loop is created to continue transcription. Otherwise, the amino acid residue would restrict the downstream motif of the stem–loop I binding to the acceptor end of charged tRNA and form a terminator loop instead, switching off the pathway of mRNA transcription (Figure 12) [77]. The initial screening method uses steady-state fluorescence-monitored ligand-binding assay and the T-box riboswitch fluorescence anisotropy assay to identify initial hit compounds. Change in fluorescence intensity indicates that the compound has hit the anti-terminator RNA, and the higher molecular weight of the tRNA–RNA complex has a more considerable anisotropy value than unbonded counterparts. The secondary screening further verifies the initial hit compounds; a fluorescence-quenching-monitored thermal denaturation assay is useful for evaluating the anti-terminator elements’ stabilization [78]. Collectively, antibacterial drug discovery targeting non-coding RNA is more approachable with this two-step screening protocol.

### 4.2. Fast Integrated Nuclease Detection in Tandem (FIND-IT)

Along with the advent of severe acute respiratory syndrome coronavirus 2, also known as SARS-CoV-2, the virus that causes COVID-19, rapid and accurate testing has been vital in slowing its spread. The current gold standard of testing utilizes quantitative reverse transcriptase polymerase chain reaction (qRT-PCR). The basic idea of this method is that viral RNA will be repeatedly copied and duplicated. Fluorescent dyes are added during the copying process, allowing for the amplification and visual detection of viral RNA. The test is extremely accurate, being able to detect just one copy of viral RNA per μL. However, the downside of qRT-PCR is that it requires specialized equipment and, therefore, a centralized lab facility. The run time for the reaction is also several hours. This is why results typically take 1–2 days after collected samples are sent to a lab to be completed.

To address these challenges, a clustered regularly interspaced short palindromic repeat (CRISPR)-based technology called Fast Integrated Nuclease Detection In Tandem (FIND-IT) was developed by the Doudna group [79]. As Figure 13 suggests, FIND-IT detects viral SARS-CoV-2 RNA by utilizing two CRISPR enzymes, type III Cas13 and type VI Csm6, at the same time. As with any CRISPR system, FIND-IT also relies on a guide RNA to prime and direct the activity of the enzymes. For this test, SARS-CoV-2 RNA serves as the guide RNA. This means that the guide RNA is complementary to a target sequence, part of the viral genome. Upon recognition of the target sequence, the activity of the enzyme is induced. In the case of Cas13, recognition of the target sequence induces it to cut an activator bound to an oligonucleotide. The activator then binds to Csm6, which induces it to cleave a dye–quencher pair, releasing a fluorescent molecule which can be visually detected. Using this method, a viral load of 30 copies/μL is detectable in as little as 20 min and always under an hour. Furthermore, the assay is able to be stored in a small, portable detector chip, and the reaction can be run at 37 °C, so immediate on-site testing is possible without extreme heat, specialized equipment, or long run times as in the case of qRT-PCR.

The tandem use of enzymes is necessary because each enzyme on its own either exhibits reduced efficiency or is unable to produce a detectable signal. For example, when Cas13 is paired to cleave fluorescent dye–quencher pairs, it takes several hours to reach a detectable signal. As for Csm6, recognition of the target sequence normally triggers the synthesis of cyclic tetra- or hexa-adenylates to serve as activators, which are rapidly degraded by the same Csm6. This means that the amount of fluorescent signal Csm6 can generate is limited. Instead, by using both enzymes in sequence, Cas13 is able to cleave a chemically modified activator which is not degraded by Csm6. As a direct result, Csm6 can cleave an indefinite number of dye–quencher pairs, leading to a large, detectable signal from a relatively small amount of substrate. The significance of this assay is that no initial amplification of the viral RNA is necessary, allowing relatively low amounts to be accurately detected. The potential of this technology is immense because, depending on the guide RNA is used, the assay could be programmed to detect a wide array of RNA species associated with different diseases [79].

## 5. Conclusions

Overall, this paper summarizes a few developing assays and tools regarding fluorescent probes in combination with RNA, some principles and explanations concerning fluorescence spectroscopy, and the potency and capacities of fluorescence assays in the development of RNA therapeutics in the past few years. However, there are still two significant barriers that need to be overcome. First and foremost, many fluorescence methods are aimed at fixed cells and living cells. The results related to dynamic RNA activities are not as convincing in living organisms. Future efforts could be concentrated on in situ RNA imaging of living animals. Secondly, fluorescence targeting RNA therapeutics are mainly focused on liver cells. In combination with RNA research, an advanced fluorescence assay should be conducted to target cancer or tumor cells in other organs.

## Figures and Tables

**Figure 1 genes-13-01348-f001:**
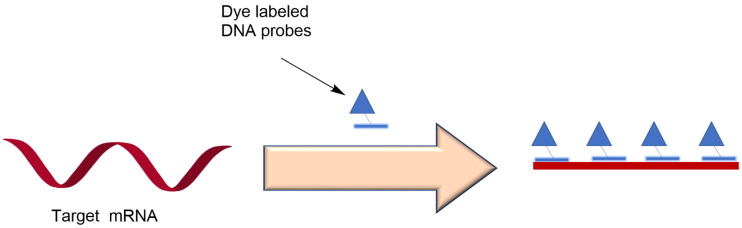
Hybridization scheme of single molecule FISH (smFISH).

**Figure 2 genes-13-01348-f002:**
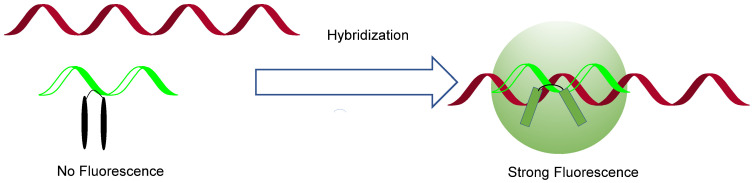
Schematic illustration of exciton-controlled hybridization-sensitive fluorescent oligonucleotide (ECHO) probes. Red stands for target RNA, and green is the ECHO probe.

**Figure 3 genes-13-01348-f003:**
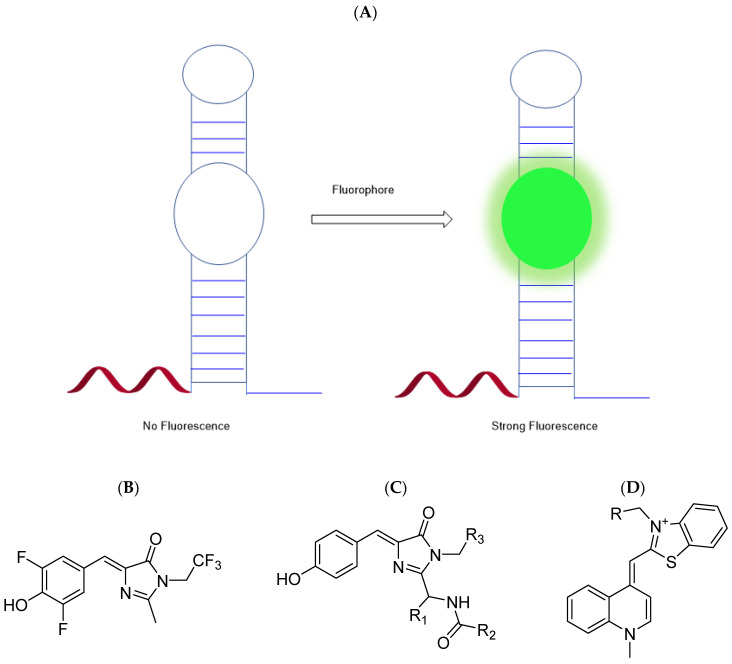
(**A**) Scheme of the aptamer–RNA complex lit up by a fluorophore. Typically, Broccoli binds to DFHBI–1T (GFP fluorophore mimic) and Mango uses TO1–biotin to activate fluorescence. (**B**) Structure of DFHBI–1T. (**C**) Structure of GFP fluorophore (for comparison). Since the flourophore is part of the larger GFP protein complex, R1, R2, and R3 correspond to serine 65, a methyl group, and glycine67 residues from the protein respectively. (**D**) Structure of TO1–biotin. R represents the biotin-polyethyleneglycol (PEG)-linker [25,26].

**Figure 4 genes-13-01348-f004:**
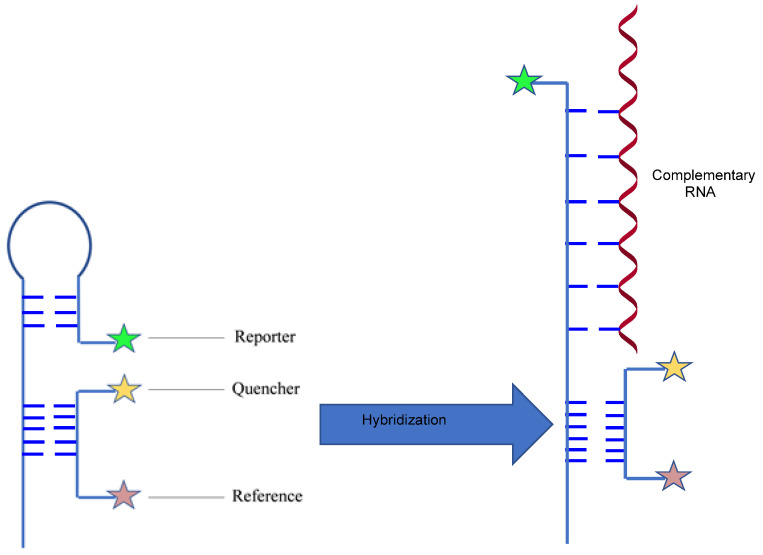
Schematic illustration of ratiometric bimolecular beacons (RBMB). It is initially in a weak fluorescence state because the quencher and reporter are close to each other; once the molecular beacon pairs with its complementary RNA, the reporter and quencher are separated, thus activating strong fluorescence.

**Figure 5 genes-13-01348-f005:**
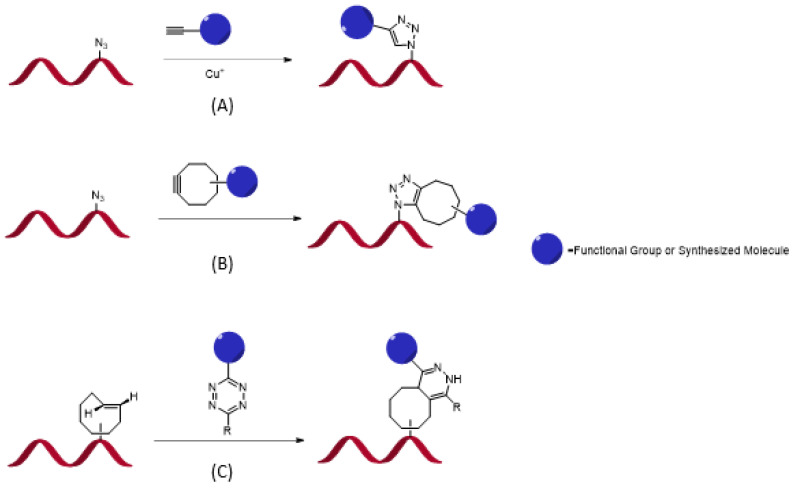
Schematic representation of RNA click reactions; (**A**)copper(I)-catalyzed azide–alkyne cycloaddition (CuAAC), (**B**) strain-promoted azide–alkyne cycloaddition (SPAAC), (**C**) inverse electron-demand Diels–Alder cycloaddition (IEDDA).

**Figure 6 genes-13-01348-f006:**
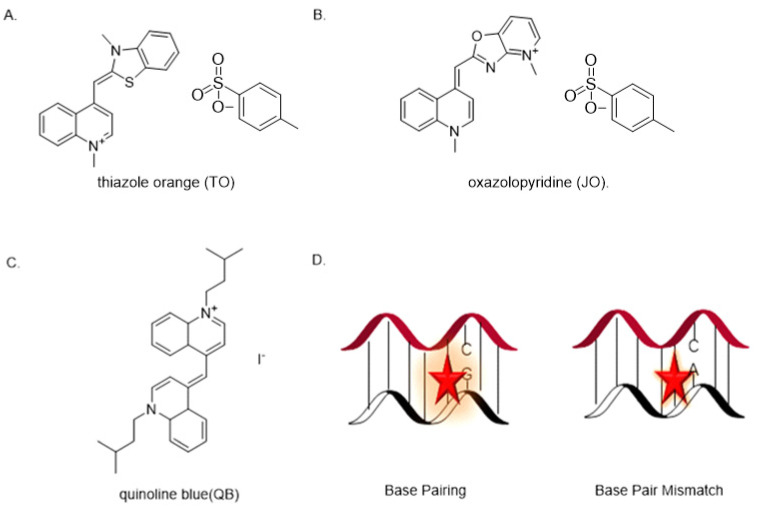
(**A**) Chemical Structure of thiazole orange (TO). (**B**) structure of oxazolopyridine (JO). (**C**) structure of quinoline blue (QB). (**D**) DNA–mRNA match and mismatch; the regular base pairing has far more brightness than the mismatch.

**Figure 7 genes-13-01348-f007:**
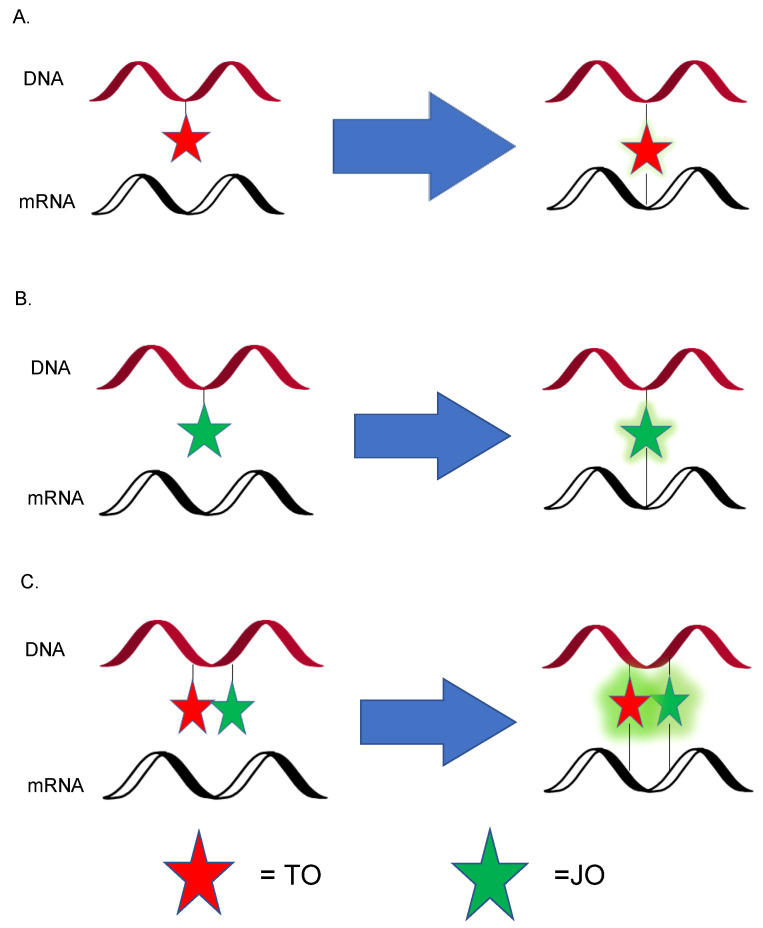
(**A**) TO labeled forced intercalation (FIT) probe. (**B**) JO labeled FIT probe. (**C**) TO–JO complex labeled FIT probe.

**Figure 8 genes-13-01348-f008:**
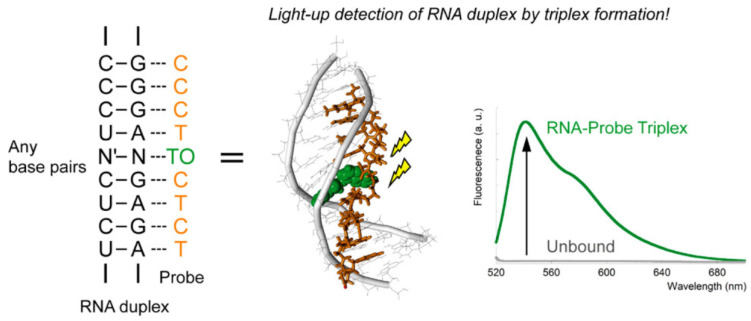
Binding of single-stranded peptide nucleic acid (ssPNA) TO modified probe to an RNA duplex motif forms a selectively fluorescent triple helix (triplex) structure.

**Figure 9 genes-13-01348-f009:**
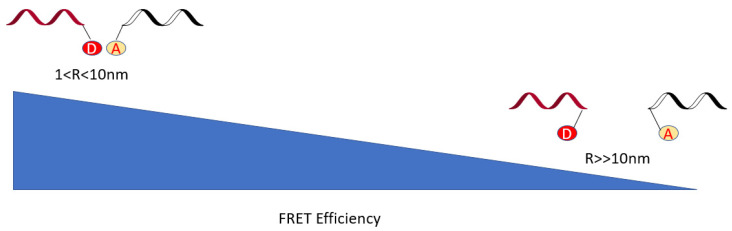
The relationship between donor and acceptor dye distances and the efficiency of fluorescence resonance energy transfer (FRET). Theoretically, the quantum yield and FRET efficiency are lower when the dyes are farther apart.

**Figure 10 genes-13-01348-f010:**
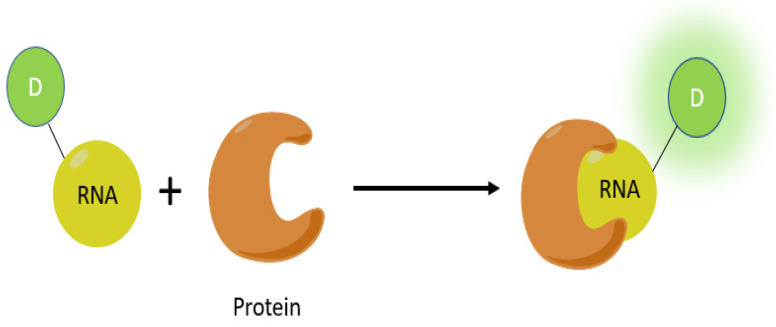
Schematic illustration of PIFE. Strong fluorescence would be observed only if the RNA binds with the target protein.

**Figure 11 genes-13-01348-f011:**
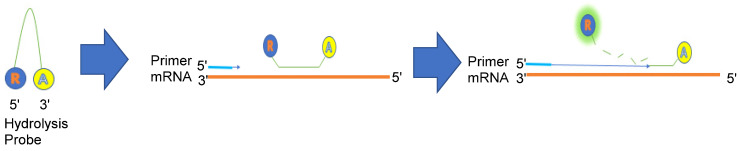
Schematic illustration of oligonucleotide hydrolysis probe when it was used for visualizing mRNA.

**Figure 12 genes-13-01348-f012:**
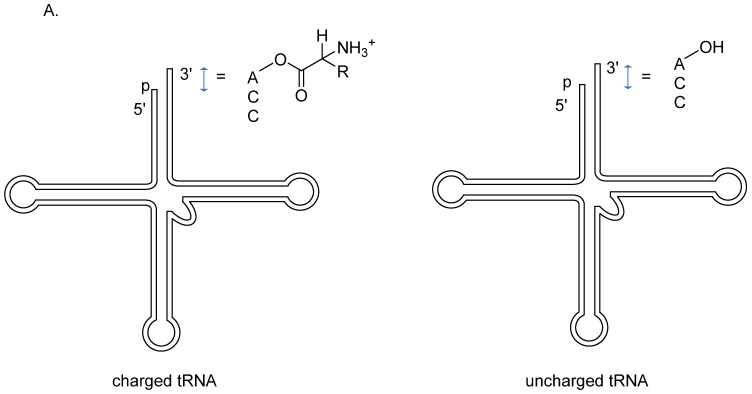
Structure of T-box riboswitch. (**A**) Both the charged and the uncharged tRNA has the CCA tail at the 3′ end. However, charged tRNA has an aminoacyl group rather than a hydroxyl group in uncharged tRNA. (**B**) Abundant of charged tRNA binds downstream of the riboswitch, stopping it from transcription. Conversely, uncharged tRNA binds with the T-box anti T/S region, and the conformational change upstream of the riboswitch promotes the formation of anti-terminator and proceeds with the transcription.

**Figure 13 genes-13-01348-f013:**
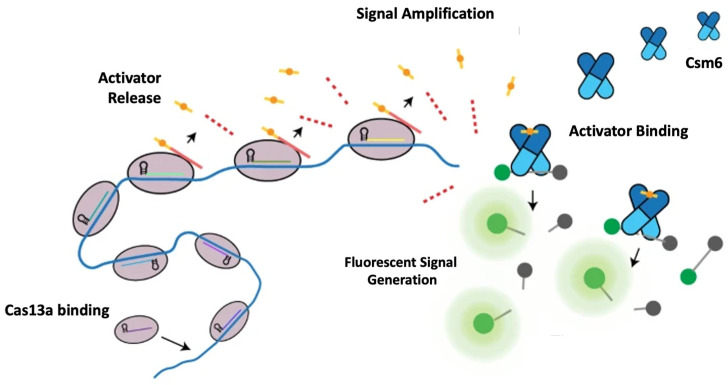
Schematic of FIND-IT assay. Cas13 binds to target RNA, inducing activator release. The activator then binds to Csm6, which goes on to release large amounts of flourescent molecules for detection [79].

## Data Availability

Not applicable.

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
