# Peer review of "Fluorescent Platforms for RNA Chemical Biology Research"

_genes, 2022, doi:10.3390/genes13081348_

Round 1

Reviewer 1 Report

Dear Author, 

It is a vey good review on RNA Biology Research. 

Please find the minor comments.

First sentence – wrong; RNA stores genetic information in viruses not in all organisms.

Second sentence can be broken down to two sentences

Clarify third sentence – regulatory mechanism is different in both of them. Add one sentence for both of them with respect to regulation.

You can make a list of type of RNAs and their roles in table.

add a note on Phosphorescence

Fluorescence in situ hybridization (FISH)

fluorescently labeled DNA or RNA probes which that is correct  can hybridize

high specificity to some (not needed) complementary target sequences

Rewrite sentence – These probes are labeled either through radiolabeling, some attached fluorescent protein, or other methods.

instead of pushed use  opened the gates of the field to a new age of study

Its widespread use has allowed it to mature and become useful for various applications - use this one - Its widespread use of this technology has allowed many improvements  and become useful for various applications.

With some current breakthrough, the technique has been largely improved – sentence can be deleted

HER2 – italics

Replace code with encode

These new technology combinations has – have

Combining fluorescence in situ hybridization and high-resolution microscopy has allowed for detecting subcellular localization- of RNA; localized RNAs.

They are found at vary levels, so to effectively detect these molecules, background fluorescence levels need to be low to get precise imaging - rewrite sentence.

from transcription in the nucleus to maturation – from transcription to maturation in the nucleus and decay in cytoplasm - rewrite like this .

Drosophila melanogaster – italics

38 reference in brackets

btuB gene italics

Sorokina et al. (year) proved this theory by binding the T7 RNA polymerase to the DNA (65)

Figure 9. schematic capital S.

Overall, I enjoyed reading this work on RNA Biology. 

Author Response

Thanks for pointing all these errors. We have corrected all the points in the revised manuscript.

Reviewer 2 Report

The review by Du et al. is well written in general. I wish that the below concerns can be addressed

Major comments:

Figure 3, Fix the panel labeling. For the structure of TO1-biotin, the structure may better be drawn as a trans conformation (see Figure 6A). State what R1, R2, R3, and R represent.

In the subsection 2.4 or in a new subsection, add a brief discussion about fluorescent detection of dsRNAs (J Am Chem Soc, 2016, 138, 9397; Biopolymers, 2022, 113, e23476; Anal Chem, 2019, 91, 5331-5338).

Minor comments:

Figure 8, FRET frequency should be FRET efficiency. 

Author Response

Thanks for the comments. We have updated the Fig. 3; added the new section of fluorescent detection of dsRNAs ; and corrected the Fig.9 in the revision.

Reviewer 3 Report

This manuscript from Du, Dartawan et al., is a review of fluorescence-based techniques applied in RNA biology. The authors do a comprehensive review of techniques; however, the manuscript itself is very hard to follow due to numerous scientific and grammatical errors. It requires extensive editing both in terms of writing and scientific content before it can be published.

Major Comments:

“Sometimes the aptamers activate fluorophores directly, and other times they can bind with GFP first. GFP is a type of gene that codes for green fluorescent proteins (22). GFP mimics spinach, broccoli, and pepper RNA. They bind and activate the fluorophores. To obtain fluorescence, aptamers usually require magnesium to activate; the amount needed depends on the exact aptamer. For example, broccoli RNA aptamer requires relatively little magnesium to ob-tain fluorescence. GFP was designed to mimic the RNA inside living cells because of the challenges of imaging RNA from the inside. Figure 3”

As written, it sounds like binding occurs to the gene not the protein. Maybe re-word this to make it less confusing. Also, GFP, while highly modified, occurs in nature and its wrong to say it mimics synthetic RNA aptamers. GFP was not designed to mimic the RNA inside living cells. I don’t understand what they are trying to say at the end here.

Also, Figure 3 labelling is awkward, as the labeling is messy (unclear what the panels are) and goes right to left at the bottom. Panel C is supposed to be the structure of GFP, but I don’t see GFP anywhere on this figure.

RNA targeting small fluorophore molecules also provide the tools to study the pro-cesses by which small drug-like organic compounds bind to specific regions of RNA to adjust its degradation, translation, and localization. There is a very brief list of small mol-ecules that target RNA and are approved by the FDA. Since these small molecules can alter the targeted RNA's conformation and localization, it is difficult to label or keep track of them for imaging purpose in live cells. Mango technology is a remote sensing image mass data storage (28). This technology allows for advanced RNA labeling and tracking with little background noise and high sensitivity to single molecules, leading the research field to the next level. “

It is unclear why this section was added to the end of the aptamer section.

“Since MBs are made of negatively charged RNA/DNA oligonucleotides, the cell membrane is generally impermeable to them.”

This implies this problem is unique to MBs, but almost all the nucleic acid based platforms discussed here suffer from the same issues.

“...donor, it relaxes from the ground state to the excited singlet state and releases some energy to get back to the ground state.”

Seems to have the process in reverse.

“Northern blotting is more accurate than qPCR.”

I’m not sure what they mean by Northern blotting being more accurate than qPCR. Accurate how?

“Fluorescence assay also makes a big difference in medicinal chemistry. Novel RNA therapies are appealing but hard to uptake into the cell. Researchers believed that natural mediums, instead of the artificial ones, could be a great tool to carry out intercellular com-munication since they belong to the target organism and there is no need to adjust the medium to a pseudo-environment. The extracellular vesicles (EV) have been recognized as good drug delivery vehicles. Jong et al. observed the process of fluorescently labeled EVs entering the Hela cell. They believe that EV uptake is caused by micropinocytosis rather than clathrin-mediated endocytosis (69).Though EV shows excellent uptake effi-ciency, it is impossible to implement batch production. Scientists discovered that human red blood cell (RBC) EV can be adopted as a strong candidate to resolve this dilemma. It could be gained because of two main reasons: first, RBCs are readily available as they are numerous in the body; secondly, RBCs do not have any nuclear or mitochondrial DNA, thus reducing the risk of horizontal gene transfer.”

The first section here is very hard to follow. Its written very confusingly. Overall, this bit is too long and focused on this finding rather than the platform used in the assay. I recommend refocusing on the platform and rewriting this section.

Minor points:

Many grammatical problems throughout. I’ll give a partial list below:

“With some current breakthrough”

The first probes were extensive since”

“These new technology combinations has greatly expanded”

RNA makes little RNA degradation negligible”

“The function of antisense transcription, COOLAIR,”

"RNA samples emits fluorescence” – fluorescence can’t be emitted

Figure 2 has a blue mark underneath arrow

“The ECHO are several 13-50nt oligonucleotide probes with”

“As one of the most developing aptamers, Mango,”

“RNA, the hair-spin stem-loop”

“and its performance in gene expression.” – roles in regulation of gene expression?

“by HCR technology (35).” – define HCR

“high quantum field are needed”

“diagnose cancer in vivo38.”

mRNA and capable of visualizing mRNPs”

upon bounding state can”

FRET is the dyes selection”

“The translation could be surpassed,” – suppressed?

the triplex-helix hydrogel is” – triple-helix?

“It located in upstream mRNA”

“t-RNA does"

Author Response

Thanks for the great comments. We have updated the manuscripts with the following six major points and a series of grammatical corrections.

1. The first major comment was in regards to the first part of section 2.2, titled “RNA-Aptamer based fluorescent assays.” The reviewer pointed out several wording and comprehension issues in this section. We largely rewrote this section to improve clarity and accuracy. In the first paragraph of 2.2, we distinguished “target” and “fluorophore.” The “target” refers to the target molecule of interest that researchers may try to highlight in a cell, while the fluorophore is the marker that associates with the aptamer to highlight dynamic changes to the target. We clarified that the process SELEX binds potential aptamers to fluorophores. For the second paragraph, we deleted the sentence regarding the GFP gene. As the reviewer stated, GFP was not designed to mimic RNA in living cells. We clarified that certain aptamers (called Spinach, Broccoli, and Pepper), bind fluorophores that mimic the fluorophore in GFP. We revised Figure 3 so that the image under C) is labelled “GFP Fluorophore.” Furthermore, we added a few sentences that explain some advantages of RNA aptamers over traditional GFP as we described how certain aptamers would bind to GFP fluorophore mimics. Finally, we deleted the last paragraph of 2.2, as MS2 is not an RNA aptamer technique, but rather more similar to an extension of the GFP technique, wherein the MS2 protein binds to a target 19mer region.

2. The reviewer’s second point was that the last paragraph of section 2.2 did not belong. We agreed, and have since decided to delete this whole paragraph.

3. The third point addresses the wording of a sentence in the second paragraph of 2.3, titled “Molecular beacon-based fluorescence assays.”  As the reviewer pointed out, the issue of MBs being negatively charged, and thus unable to cross the cell membrane, is not unique. We addressed the fact that other platforms discussed in the paper also have this issue.

4. The fourth point made by the reviewer suggests that we described the process of FRET backwards. We revised this section to make it clear that the donor is excited, and then upon returning to ground state, transfers its energy to the acceptor, resulting in acceptor fluorescence.

5. The fifth point we addressed by removing the statement that Northern blotting is more accurate than qPCR, as it is too general and inaccurate.

6. To address the sixth point about section 4.1, titled “Fluorescence techniques in RNA therapeutics,” we rewrote to more focus on the fluorescence techniques used to aid studies in the advancement of RNA drugs.

7. Finally, the reviewer pointed out several grammatical issues. We have gone through the manuscript and corrected these errors, as well as some other grammatical/formatting issues that we found throughout.